# Semantic Activation in Badminton Action Processing and Its Modulation by Action Duration: An ERP Study

**DOI:** 10.3390/brainsci12111458

**Published:** 2022-10-27

**Authors:** Ruohan Chang, Xiaoting Wang, Jinfeng Ding

**Affiliations:** 1School of Psychology, Beijing Language and Culture University, Beijing 100083, China; 2School of Psychology, Beijing Sport University, Beijing 100084, China; 3CAS Key Laboratory of Behavioral Science, Institute of Psychology, Beijing 100101, China; 4Department of Psychology, University of Chinese Academy of Sciences, Beijing 100049, China

**Keywords:** semantic activation, action processing, P300, LPC, badminton

## Abstract

Action processing is crucial for sports activities. Using event-related potentials (ERPs), the present study investigated whether semantics were activated in action processing and, if so, whether semantic activation was modulated by action duration. Badminton athletes were recruited to finish a lexical decision task following an action-semantic priming paradigm, in which short (400 ms) or long (1000 ms) action videos served as primes, and semantically congruent or incongruent action words served as targets. The ERP results showed a P300 effect, that is, larger P300 amplitudes were observed for targets primed by semantically incongruent action videos than for targets primed by semantically congruent action videos, only when the action videos were long and not when the action videos were short. Moreover, a late positive component (LPC) was only sensitive to action duration, showing that the targets primed by long action videos elicited larger LPC amplitudes compared to the targets primed by short action videos. These results suggested that semantics could be activated in action processing and that semantic activation was modulated by action duration, supporting a link between the language system and action processing.

## 1. Introduction

Increasing evidence has claimed that there are strong links between language and the motor system [1,2]. It has been demonstrated that the sensory-motor system plays a functional role in action-related language processing, the so-called embodied language comprehension theory [3,4]. According to this theory, semantic processing of action-related linguistic stimuli is, at least in part, grounded in brain areas that generally subserve perception and action [5]. Ample evidence has shown that sensory and motor regions involved in perception and action are activated during the semantic processing of action-related language stimuli [6,7]. For example, functional magnetic resonance imaging (fMRI) studies have found that reading action words, such as “lick”, “pick”, or “kick”, simulates brain areas that are also involved in actual movement of the tongue, fingers, or feet, respectively [8]. Moreover, behavioural and neurophysiological evidence has indicated that modifying activities in the sensory-motor system influence action-related language processing [9,10,11]. For example, it has been found that both short-term motor behaviours and long-term motor experience modulate action-related semantic processing [11]. Furthermore, transcranial magnetic stimulation (TMS) studies have shown that magnetic pulse stimulation of the motor system influenced language task performance of action-related words [12]. This evidence provides support for the activation and modulation of the motor system in action-related semantic processing. However, relatively less is known about whether the language system is activated in action processing.

Action processing is a key component of human cognition and is especially important for excellent performance in sports activities [13]. An observed action could be processed at multiple levels, including both mapping kinematic and motor information onto the concrete representation level and inferring the conceptual goal and intention at the abstract representation level [14,15,16]. Using varying levels of temporal occlusion as one of the experimental manipulations to action videos, numerous psychophysical and neurophysiological studies have investigated action processing at the concrete level and found that action video durations influenced action anticipation [17,18,19,20]. In addition, elite athletes could extract kinematic and motor information to predict the action outcome (e.g., where the shuttlecock would land) earlier and more accurately than people who have no direct motor experience [17,18,19,20]. This superior anticipation ability is believed to mainly rely on a brain network known as the action mirror neuron system (MNS) or action-observation network (AON) [13,18].

Recently, several studies have focused on action processing at the abstract level and found that the language system might also contribute to action processing [21,22]. For example, in a fMRI study, Wang and colleagues (2019) found that beyond the sensorimotor regions, some semantic regions, such as the posterior inferior parietal lobe, middle temporal gyrus, and ventromedial prefrontal cortex, were also engaged in action anticipation, indicating that action anticipation might also engage conceptual level analysis beyond the sensorimotor level. In further research, behaviour and neural results showed that the memory of hand-related action verbs impaired the accuracy of concurrent action anticipation in table tennis athletes, indicating that language and action processing might share resources and that effector-specific linguistic semantics interfere with action anticipation [21,22]. These studies are valuable for revealing action processing at the abstract level and indicate that action anticipation might involve conceptual level analysis and be modulated by semantic processing. However, the question of whether athletes can predict action semantics at the abstract level in action processing still requires investigation. The present study aimed to investigate this question.

Semantic priming has served as an important tool in studies exploring semantic processing in nonlinguistic stimuli, such as sounds [23], music [24], and odour [25]. Overall, those studies have shown a semantic priming effect, that is, a reduced N400 amplitude for semantically congruent/related targets compared to that observed in response to semantically incongruent/unrelated targets [23,24,25]. The N400 component is a broad negative deflection of the event-related potential (ERP), and the N400 effect in semantic priming has been considered an index of semantic processing of nonlinguistic stimuli [24,26]. In addition, the P300 component has been observed in semantic priming, with increased amplitudes evoked by semantically unrelated targets compared to semantically related targets [27,28]. The P300 component has been observed in various tasks that require stimulus discrimination, with low-probability target items eliciting higher amplitudes than high-probability nontarget items [29]. In the semantic priming paradigm, the P300 component has been considered to reflect access to semantic memory [27,28]. The higher P300 amplitudes in response to semantically unrelated targets indicated that accessing the semantic memory of semantically unrelated targets requires more cognitive effort compared to that needed for semantically related targets [27,28]. In addition, a late positive component (LPC) is reported to be sensitive to semantic priming, with larger amplitudes observed in responses to target words that are preceded by semantically related words than to target words that are preceded by semantically unrelated words, reflecting the semantic relationship detection process [30].

Following previous studies [23,24,25], a cross-domain semantic priming paradigm was used in the present study to investigate semantic activation in action processing, with action video as primes and semantically congruent or incongruent action words as targets. Moreover, given that the duration of an action video would influence action processing, that is, a long video provides more action information and allows more elaborative analysis compared to a short video [17,31], we also included action video duration as a factor to investigate whether and how it modulates semantic activation in action processing.

Overall, the present study aimed to investigate whether action semantics were predicted/activated during badminton action processing and, if so, whether and how semantic activation was modulated by action duration. Badminton athletes were instructed to finish a cross-domain semantic priming procedure followed by a lexical decision task, in which a long (1000 ms) or short (400 ms) action video served as the prime and a conceptual congruent or incongruent action word served as the target, and participants were asked to decide whether the target stimulus was a real word. We hypothesized that if the action concept could be activated during action processing, then the semantic priming effect would be observed, as indicated by an N400, P300, and/or LPC effect. In contrast, if the semantic concept could not be activated during action processing, then the semantic priming effect would not be observed. In addition, given that a long action video might provide more information than a short action video, we hypothesized that the semantic activation might be stronger when the action video is long than when the action video is short, as reflected by a stronger semantic priming effect for videos with long durations than for those with short durations.

## 2. Materials and Methods

### 2.1. Participants

Twenty badminton players (15 males, mean age = 19.6 years; range = 18.0–25.0 years; SD = 1.6 years) participated in this study. They were qualified as the second level or above the level of the national standard (1 at the second level, 17 at the first level, and 2 at the elite level). They had a mean of 9.6 years of training experience (range = 6.0–15.0 years; SD = 2.4 years) and practised badminton at least twice a week, 2 h each time, in the last 2 years. All participants were right-handed with normal or correct-to-normal vision. They were native speakers of Mandarin Chinese and did not report any neurological disorders. Informed consent was obtained prior to the experiment. The data of three male participants were excluded from the statistical analysis, two due to partly missing data and another due to excessive artefacts.

### 2.2. Materials

We used 120 video–word pairs as experimental materials in a lexical decision task. In each pair, a video served as a prime stimulus, and an action word served as a target stimulus. The videos were silent colour clips (AVI format, 30 frames per second), selected from the 2018 World Badminton Championship and the last eight of the finals at the end of that year. Kinovea software was used to capture 1000 ms and 400 ms of the video up to the point of 40 ms before the contact between the shuttlecock and the racket [32]. Each duration condition consisted of 60 videos, including 20 clears, 20 drops, and 20 smashes. The action words (“高球”, clear; “吊球”, drop; and “杀球”, smash) were used as target stimuli and were manipulated to be semantically congruent or incongruent with the prime videos. For each duration condition, we used another sixty video-pseudoword pairs as filler materials, with action videos serving as prime stimuli and pseudowords serving as target stimuli. Each pseudoword consists of two real Chinese characters whose combination lacks meaning in Mandarin Chinese. These fillers were used to mask the experimental purpose and counterbalance the response in a lexical decision task.

To match the processing difficulty of the videos among the four conditions, seven badminton players (5 males, mean age = 23.1 years; range = 19.0–28.0 years; SD = 3.0 years; with a mean of 7.7 years training experience, range = 5.0–14.0 years, SD = 3.3 years) who were qualified as the second level or above the level of the national standard were recruited to watch the videos and judge their action concepts by pressing “J”, “K”, or “L” to represent clear, drop, or smash, respectively, as quickly and accurately as possible. These players did not participate in the electroencephalogram (EEG) experiment. Two-way repeated measures analysis of variances (ANOVA) was performed with action duration (long or short) and semantical congruence (congruent or incongruent) as two factors, and the results did not reveal any significant effect of either accuracy (Fs ≤ 0.04, *ps* ≥ 0.844; M ± SD: long–congruent, 85.71 ± 11.26%; long–incongruent, 85.71 ± 11.26%; short–congruent: 85.24 ± 15.23%; short–incongruent: 85.24 ± 15.23%) or reaction time (Fs ≤ 1.03, *ps* ≥ 0.318; M ± SD: long–congruent, 754.24 ± 172.66 ms; long–incongruent, 760.33 ± 167.57 ms; short–congruent: 752.75 ± 130.43 ms; short–incongruent: 707.54 ± 158.82 ms). These results indicated that the processing difficulty of the action videos was well matched among the four conditions.

### 2.3. Procedure

Participants were comfortably seated in front of a screen, approximately 60 cm away. For each duration condition, the 120 stimulus pairs (60 experimental materials and 60 filler materials) were presented in a pseudorandom order with no more than three consecutive trials under the same conditions. The 120 stimulus pairs were presented in two blocks, and each block consisted of 60 trials. In each trial, a white fixation cross was presented in the centre of a black background for 1000 ms. Then, a 1000 ms or 400 ms prime video was presented. Following a 100 ms black screen, the target stimuli were presented in white on a black background for 300 ms. Participants were instructed to watch the video and to judge whether the target stimulus was a word or not as quickly and accurately as possible (see Figure 1 for the experimental procedure). Half of the participants were asked to press “F” or “J” with their index fingers to indicate a word or a pseudoword, respectively, while the buttons indicated the opposite responses for the other half of the participants. Moreover, half of the participants finished the long duration condition first and then the short duration condition, while the other half completed the tasks in a reverse order. Participants performed 12 practice trials and then completed all four blocks, with brief rest periods separating the blocks.

### 2.4. EEG Recording and Preprocessing

EEG data were recorded with a 64-channel Ag/AgCl electrode cap according to the Standard International 10-10 system using Curry 7.0 software. EEG data were referenced online to the left mastoid. Vertical electrooculography (EOG) was recorded from electrodes placed above and below the left eye, and horizontal EOG was recorded from electrodes placed at the left and right outer canthi. The sampling rate was 1000 Hz. The impedance was kept below 5 kΩ for all electrodes.

EEG data were preprocessed using EEGLAB [32] and ERPLAB [33] analysis packages in the MATLAB environment (The Math Works, Inc.). After importing the data for processing, the EEG data were rereferenced offline to the average of the left and right mastoid channels. Then, the sampling rate was changed to 500 Hz to reduce the data size, and a bandpass filter of 0.1–30 Hz (24 dB per octave) was applied to the continuous data. The EEG data were then segmented from 100 ms before to 1000 ms after the onset of the target stimuli, with baseline correction from 100 ms to 0 ms preceding target stimuli onset. After that, ocular blinks were corrected using independent component analysis (ICA), and artefact rejection was performed with a ± 75 μV amplitude criterion. Finally, 6.67% of the trials were rejected due to artefacts, and the remaining trials did not differ significantly across the four conditions (Fs ≤ 3.05, *ps* ≥ 0.10; M ± SD: long–congruent, 27.82 ± 1.94; long–incongruent, 27.47 ± 2.15; short–congruent, 28.12 ± 1.90; short–incongruent, 28.59 ± 1.91).

### 2.5. ERP Data Analysis

Based on visual inspection, larger P300 amplitudes were elicited by target words primed by incongruent action videos compared to those elicited by target words primed by congruent action videos in long duration conditions, and not in short duration conditions, and larger LPC amplitudes were elicited by target words primed by long action videos compared to those elicited by target words primed by short action videos. For statistical analysis, the latency windows of P300 and LPC were selected as 200–400 ms and 500–700 ms, respectively, based on previous studies [34,35] and visual inspections. For each time window, average waveforms of the ERPs for the target stimuli were computed across all remaining trials per condition for each participant. Then, a four-way repeated-measure ANOVA was conducted for the average amplitude, with duration (long or short), congruence (congruent or incongruent), hemisphere (left, medial, and right) and anteriority (frontal, central and parietal) serving as four within-subject factors. In this way, electrodes were organized into nine regions of interest (ROIs), each containing five or six representative electrodes: left frontal (F3, F5, F7, FC3, FC5, and FT7), left central (C3, C5, CP3, CP5, and TP7), left parietal (P3, P5, P7, PO5, PO7, and O1), medial frontal (F1, FZ, F2, FC1, FCZ, and FC2), medial central (C1, CZ, C2, CP1, CPZ, and CP2), medial parietal (P1, PZ, P2, PO3, POZ, and PO4), right frontal (F4, F6, F8, FC4, FC6, and FT8), right central (C4, C6, CP4, CP6, and TP8), and right parietal (P4, P6, P8, PO6, PO8, and O2). For ANOVA, when Mauchly’s test of sphericity was significant, the Greenhouse–Geisser correction was applied. All significant interactions involving action duration and semantic congruence were followed by simple effect tests.

## 3. Results

### 3.1. Behavioural Data

Mean accuracy and reaction time in the lexical decision task were computed, and the results are shown in Table 1. For accuracy, two-way repeated-measures ANOVA with duration (long or short) and congruence (congruent or incongruent) as two within-subjects factors was conducted, and the results revealed a significant interaction between duration and congruence (F (1, 16) = 6.18, *p* = 0.024, η^2^*p* = 0.28). However, follow-up simple effect analysis revealed neither a significant main effect of congruence for long or short durations (Fs ≤ 2.24, *ps* ≥ 0.154), nor a significant main effect of duration for congruent or incongruent conditions (Fs ≤ 2.52, *ps* ≥ 0.132).

To compute the mean reaction time, the data with erroneous responses (4.90% of the data) and with reaction times shorter than 300 ms or longer than 1000 ms (5.44% of the data) were removed first. Then, two-way repeated-measures ANOVA was conducted for the remaining data. The results revealed neither a significant main effect of duration (F (1, 16) = 0.01, *p* = 0.943, η^2^*p* ≤ 0.001) nor a significant main effect of congruence (F (1, 16) = 0.65, *p* = 0.434, η^2^*p* = 0.04). There was also no significant interaction between duration and congruence (F (1, 16) = 0.43, *p* = 0.521, η^2^*p* = 0.03).

Overall, the behavioural results did not reveal any significant effect involving duration and congruence factors.

### 3.2. ERP Data

Table 2 presents the repeated-measures ANOVA results in the 200–400 ms and 500–700 ms time windows. Figure 2 presents the grand average waveforms elicited by the target words primed by long–congruent, long–incongruent, short–congruent, and short–incongruent videos at nine representative electrodes (F3/FZ/F4, C3/CZ/C4, and P3/PZ/P4). Figure 3 presents the topographies of the difference waveforms in the 200–400 ms and 500–700 ms time windows.

In the 200–400 ms time window, there was a significant main effect of congruence and a significant three-way interaction of congruence × hemisphere × anteriority (see Table 2). This three-way interaction of congruence was further evaluated by conducting separate analyses for each anteriority level. For the frontal regions, an ANOVA with congruence and hemisphere as within-subject factors revealed a significant main effect of congruence (F (1, 16) = 9.66, *p* = 0.007, η^2^*p* = 0.38), showing that incongruent words elicited larger P300 amplitudes than congruent words (mean ± SE: incongruent, 5.31 ± 0.78 μV; congruent, 4.38 ± 0.78 μV; difference (incongruent–congruent), 0.93 ± 0.30 μV). For the central regions, the ANOVA results revealed a significant main effect of congruence (F (1, 16) = 4.99, *p* = 0.040, η^2^*p* = 0.24) and a significant interaction between congruence and hemisphere (F (2, 32) = 4.43, *p* = 0.020, η^2^*p* = 0.22). Follow-up simple effect analysis revealed that incongruent words elicited larger P300 amplitudes over the right hemisphere than congruent words (F (1, 16) = 7.15, *p* = 0.017, η^2^*p* = 0.31; mean ± SE: incongruent, 6.07 ± 0.87 μV; congruent, 5.37 ± 0.83 μV; difference (incongruent–congruent), 0.70 ± 0.26 μV) but not over the left and medial electrodes (Fs ≤ 4.16, *ps* ≥ 0.058). For the parietal regions, the ANOVA results revealed neither a significant main effect of congruence nor a significant interaction between congruence and hemisphere (Fs ≤ 2.89, *ps* ≥ 0.109). Overall, a congruence effect was observed only over the frontal and right central regions.

Moreover, repeated-measures ANOVA revealed a significant interaction between duration and hemisphere (see Table 2). The follow-up simple effect analysis revealed that the long condition elicited larger P300 amplitudes over the right hemisphere than the short condition (F (1, 16) = 4.71, *p* = 0.045, η^2^*p* = 0.23; mean ± SE: long, 5.27 ± 0.88 μV; short, 3.70 ± 0.59 μV; difference [long-short], 1.57 ± 0.72 μV) but not over the left and medial electrodes (Fs ≤ 1.93, *ps* ≥ 0.184).

Most importantly, repeated-measures ANOVA revealed a significant three-way interaction of duration × congruence × anteriority (see Table 2). This three-way interaction was broken down by conducting separate analyses for each duration level. For the long duration, an ANOVA with congruence and anteriority as the within-subject factors revealed a significant main effect of congruence (F (1, 16) = 11.95, *p* = 0.003, η^2^*p* = 0.43) and a significant interaction between congruence and anteriority (F (2, 32) = 6.57, *p* = 0.017, η^2^*p* = 0.29). The follow-up simple effect analysis revealed that, compared to congruent words, incongruent words elicited larger P300 amplitudes over all three anteriority levels (frontal: F (1, 16) = 14.34, *p* = 0.002, η^2^*p* = 0.47; mean ± SE: incongruent, 6.10 ± 1.15 μV; congruent, 4.24 ± 1.10 μV; difference (incongruent–congruent), 1.87 ± 0.49 μV; central: F (1, 16) = 8.20, *p* = 0.011, η^2^*p* = 0.34; mean ± SE: incongruent, 5.94 ± 0.89 μV; congruent, 4.96 ± 0.81 μV; difference (incongruent–congruent), 0.96 ± 0.34 μV; parietal: F (1, 16) = 7.53, *p* = 0.014, η^2^*p* = 0.32; mean ± SE: incongruent, 4.20 ± 0.71 μV; congruent, 3.22 ± 0.66 μV; difference (incongruent–congruent), 0.98 ± 0.36 μV). For the short duration, the ANOVA revealed neither a significant main effect of congruence nor a significant interaction between congruence and anteriority (Fs ≤ 0.10, *ps* ≥ 0.832), suggesting that no P300 effect was present in the short condition. Overall, a P300 effect in response to congruence was observed only in the long condition, not in the short condition.

In the 500–700 ms time window, repeated-measures ANOVA revealed a significant main effect of duration (see Table 2), showing that the long condition elicited larger LPC amplitudes than the short condition (mean ± SE: long, 10.68 ± 1.45 μV; short, 6.23 ± 1.29 μV; difference (long–short), 4.44 ± 1.81 μV).

Overall, the ERP results revealed a congruence effect for the long duration condition but not for the short duration condition in the 200–400 ms time window and showed a duration effect for both the congruent and incongruent conditions in the 500–700 ms time window.

## 4. Discussion

The human conceptual system encapsulates people’s knowledge about the world and supports a wide variety of cognitive operations, including attention, memory, language, thought, and sociocultural cognition [36]. The present study aimed to investigate whether semantic concepts could be activated in action processing and, if so, whether semantic activation was modulated by action duration. A modified version of the semantic priming paradigm was used, with long or short action videos as primes and semantically congruent or incongruent action words as targets, followed by a lexical decision task. The behavioural results did not reveal any significant effects. The ERP results showed a larger P300 for target words primed by incongruent action videos compared to target words primed by congruent action videos in the long duration condition but not in the short duration condition. Moreover, the LPC was sensitive to action duration, showing larger amplitudes for the long condition than for the short condition, but was insensitive to action-semantic congruence.

### 4.1. Semantic Activation in Action Processing

Following previous studies investigating conceptual processing in nonlinguistic systems [23,24,25], we used a cross-domain priming paradigm to investigate semantic activation in action processing. The behavioural results of the lexical decision task did not reveal a semantic priming effect. This is inconsistent with some previous studies of cross-domain priming, which reported shorter reaction times in semantically congruent/related pairs than in semantically incongruent/unrelated pairs [23,37]. However, it should be pointed out that the priming effect at the behavioural level is not a general finding. For example, in a series of carefully controlled experiments using a naming task with word targets, Klauer and Musch (2001) did not find any evidence of an affective priming effect at the behavioural level [38]. In the study by Daltrozzo and Schön (2009), the behavioural results of a relatedness judgment task did not reveal any priming effect of music on word processing [24]. In the present study, the absence of a significant effect in the lexical decision task indicated that the priming effect of the action videos on word processing was not observed at the behavioural level.

Unlike the behavioural data, the ERP data revealed a semantic priming effect with a modulation of the P300 component; that is, larger P300 amplitudes were elicited by targets primed by incongruent action videos compared to targets primed by congruent action videos over a long duration. The P300 component has often been observed in the “oddball” paradigm, with low-probability target items eliciting higher amplitudes than high-probability nontarget items, and has been linked to attention and subsequent memory processing [29]. In the semantic priming paradigm, the P300 has been found to be sensitive to semantic relatedness, with unrelated targets eliciting higher amplitudes than related targets, and has been considered to reflect access to semantic memory [27,28]. Accepting this view, larger P300 amplitudes elicited by incongruent targets compared to congruent targets in the present study might indicate that accessing semantic memory of incongruent targets was associated with more cognitive effort, whereas accessing semantic memory of congruent targets could benefit from the semantic prediction and activation of action videos. Therefore, the semantic priming effect in our data could be an index of semantic activation in action processing and could provide a new piece of evidence for the links between language system and action processing.

One might argue that there was a reverse N400 rather than a P300 effect in the early time window, that is, more negative-going waveforms were elicited by congruent targets compared to incongruent targets. Actually, a reverse N400 effect has been found for the priming effect, with more negative-going waveforms elicited by related targets than by unrelated targets when the prime was masked [39] or when the prime-target interval was short [40]. The reverse N400 effects have been explained by a centre-surround inhibition mechanism, assuming that under certain circumstances (such as low visibility for visual primes and short duration for emotional prosodic primes), a prime would activate the corresponding node only weakly, and the surrounding nodes would be inhibited to sustain the relative activation difference between the centre and its surroundings [39,40]. This centre-surround inhibition mechanism serves to facilitate access to the weakly activated prime concept but also hampers access to the surrounding related concept [39]. In these studies, semantic or affective relatedness between the prime and target items was manipulated, the prime concepts were considered the centre nodes, and their related concepts were considered the surrounding nodes. In the present study, however, semantic conceptual congruency between the action video prime and word target items was manipulated. The congruent target words should be considered as the centre nodes and the incongruent targets words should be considered as the surrounding nodes. If the centre-surrounding inhibition mechanism functioned in the current study, accessing the incongruent target concepts would become more difficult. This would result in larger N400 amplitudes for the incongruent targets relative to the congruent targets, which is inconsistent with our results.

To our knowledge, this is the first study to directly investigate whether action semantics can be predicted and activated in action processing. Our results indicated that individuals can predict the upcoming actions and activate the action semantics at the abstract representation level, providing neurophysiological evidence for semantic prediction and activation in action processing. The neural correlates of this semantic prediction and activation need to be explored in future studies: specifically, how the brain utilises the visually presented action information to infer the goal and intention at the abstract representation level in action processing. Moreover, contrasting ERPs elicited by words and pseudowords matched in visual properties would reveal getting access to lexico-semantic information, which will provide an opportunity to explore the potential difference between accessing to lexico-semantic and action-semantic information.

### 4.2. Action Duration Modulated Action-Semantic Activation

The P300 effect was observed only in the long condition but not in the short condition, indicating that action duration modulates semantic activation in action processing. Action duration has been found to influence decision making in sports activities; that is, a long action video provides rich information to support athletes in comprehensively processing and making analytical decisions, whereas a short action video provides little information, and athletes can only make intuitive decisions [31]. Similarly, our results showed a superiority of long action videos compared to short action videos in semantic activation, with long action videos activating corresponding semantic concepts and influencing the processing of the following action words at an early stage, whereas short action videos did not elicit semantic activation. These results should be attributed to a larger amount of information provided by long action videos compared to short action videos. This raises an interesting question of how long of a duration is enough to activate a corresponding semantic in action processing. The minimal duration of an action video for semantic activation should be estimated by using action videos of 400 ms to 1000 ms duration in future studies.

It is worth noting that a block design was used in the present study to prevent the participants feeling the action videos disordered or guessing the presentation rules of the materials. However, the blocks with long duration videos would take more time that the participants may need to concentrate more, or they may be more engaged and paying more attention, resulting in semantic activation in long action videos condition rather than in short action videos condition. To exclude this possibility, a mixed design could be used to verify the results in the future.

Moreover, one might argue that 400 ms video clips might be quite brief and ambiguous, and in this case the actions between clear/drop/smash were not easily distinguished. This might result in the absence of the P300 effect in the short condition. However, this possibility could be ruled out given that the action videos had been pretested offline and the mean accuracy of judging the corresponding action concepts was above 85% in the short duration condition (with no difference with the long duration condition).

### 4.3. The LPC Was Sensitive to Action Duration but Not to Action-Semantic Congruence

Significant effects of action duration on the LPC component were found with enhanced amplitudes observed for long durations than for short durations. However, no effects of action-semantic congruence were found on the LPC component. In previous studies, the larger LPC in response to semantically related targets compared to unrelated targets has been linked to the semantic relationship detection process [30,41]. Inconsistent with these studies, our results showed the lack of sensitivity of the LPC to action-semantic congruence, indicating that the LPC might reflect another kind of cognitive process different from semantic relationship detection. Alternatively, the LPC has been associated with general semantic processing [42,43]. According to this interpretation, it is probably that long action videos may elicit deeper semantic processing for target words compared to short action videos, regardless of congruency effects.

Moreover, given that the LPC has been proposed to reflect the retrieval of episodic information [44,45], we tentatively interpreted the LPC results in our data in terms of episodic memory. In previous studies, a larger LPC has been observed when stimuli were associated with more episodic memories [44]. In the present study, when participants read the target words, the previously presented videos might be recollected. Relative to short action videos, long action videos provided more episodic information being retrieved, thus resulting in larger LPC amplitudes for long duration condition than for short duration condition.

## 5. Conclusions

The present study showed that the P300 and the LPC, two ERP components that index different stages of information processing, are differentially sensitive to action-semantic congruence and action duration in the action-semantic priming paradigm. Our data indicated that semantic concepts could be predicted and activated in badminton action processing and that semantic activation was modulated by action duration. These results support the strong link between linguistic system and action processing.

## Figures and Tables

**Figure 1 brainsci-12-01458-f001:**
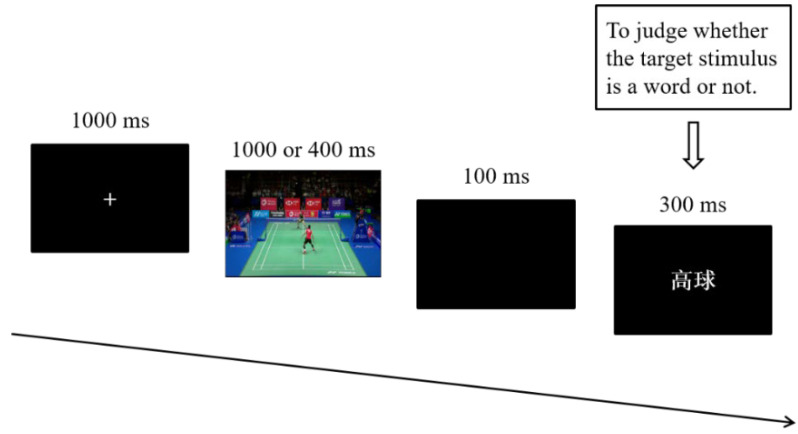
Experimental procedure in this study.

**Figure 2 brainsci-12-01458-f002:**
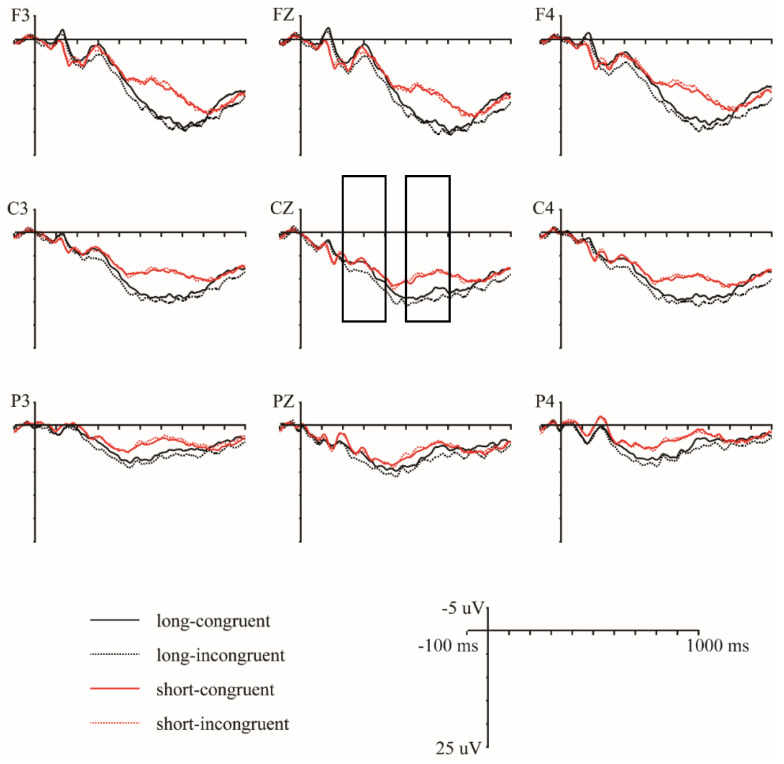
Grand average waveforms evoked by the target words primed by long–congruent, long–incongruent, short–congruent, and short–incongruent videos at nine selected electrode sites. Waveforms are time-locked to the onset of the target words, and negative amplitudes are plotted up.

**Figure 3 brainsci-12-01458-f003:**
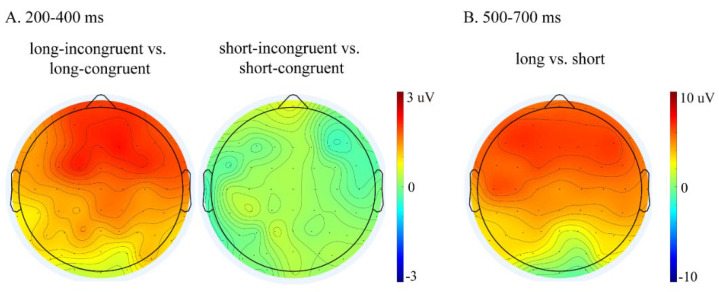
(**A**) Topographies of the difference waveforms determined by subtracting ERPs for the long–congruent conditions from those of the long–incongruent conditions (left) and by subtracting ERPs for the short–congruent conditions from those of the short–incongruent conditions (right) in the 200–400 ms time window. (**B**) Topographies of the difference waveforms calculated by subtracting ERPs for the short conditions (the average of the short–congruent and short–incongruent conditions) from those of the long conditions (the average of the long–congruent and long–incongruent conditions) in the 500–700 ms time window.

**Table 1 brainsci-12-01458-t001:** Behavioural results in the lexical decision task.

Mean (SD)	Accuracy (%)		Reaction Time (ms)	
	Congruent	Incongruent	Congruent	Incongruent
Long	93.73 (5.26)	95.69 (6.75)	434.66 (49.31)	435.07 (52.68)
Short	95.29 (5.41)	93.92 (6.79)	433.48 (45.08)	437.69 (43.45)

**Table 2 brainsci-12-01458-t002:** ERP results of repeated-measures ANOVAs.

Effect	*df*	200–400 ms			500–700 ms		
		F	*p* Value	η^2^*p*	F	*p* Value	η^2^*p*
C	1, 16	6.91 *	0.018	0.30	0.52	0.483	0.03
C × H	2, 32	3.12	0.058	0.16	0.90	0.417	0.05
C × A	2, 32	1.76	0.201	0.10	0.03	0.915	0.002
C × H × A	4, 64	2.60 *	0.044	0.14	0.07	0.950	0.004
D	1, 16	2.74	0.118	0.15	6.04 *	0.026	0.27
D × H	2, 32	4.99 *	0.013	0.24	0.03	0.972	0.002
D × A	2, 32	0.40	0.567	0.02	3.49	0.078	0.18
D × H × A	4, 64	1.32	0.282	0.08	0.31	0.759	0.02
D × C	1, 16	3.76	0.070	0.19	1.42	0.251	0.08
D × C × H	2, 32	1.36	0.270	0.08	0.19	0.829	0.01
D × C × A	2, 32	5.73 **	0.007	0.26	0.45	0.640	0.03
D × C × H × A	4, 64	2.28	0.117	0.13	1.40	0.258	0.08

Notes: C: congruence; H: hemisphere; A: anteriority; D: duration. Significant effects are marked with asterisk(s) following F values. * Significant at 0.05 level; ** Significant at 0.01 level.

## Data Availability

The datasets generated for this study are available on request to the corresponding author.

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
