# Peer review of "Semantic Activation in Badminton Action Processing and Its Modulation by Action Duration: An ERP Study"

_brainsci, 2022, doi:10.3390/brainsci12111458_

Round 1
Reviewer 1 Report
The authors proposed an ERP study about the relationship between semantic activation and action processing in badminton athletes. A modified version of the semantic priming paradigm was employed, including long or short action videos as primes and semantically congruent or incongruent action words as targets, followed by a lexical decision task. The ERP results revealed a larger P300 amplitude for targets primed by incongruent action videos compared to targets primed by congruent action videos in the long duration condition but not in the short duration condition. Moreover, the LPC showed larger amplitudes for the long condition than for the short condition, but was not affected by action-semantic congruence. The topic is very interesting, the paper is well-structured and detailed but it is not clear to me what is the novelty of this work and possible future research based on your findings. Please point them out in the paper.
Minor comments:
1) The 10-20 system includes up to 21 electrodes. Do you refer to 10-10 system? (Line 175).
2) The values marked in bold in Table 2 are not easily distinguishable, mark them better.
Reviewer 2 Report
This study sought to examine how video clips of badminton play may prime word processing of action verbs by using ERP. They found that a late positive component larger for primes with long than short videos, while a P300 congruency effect only for primes with long videos and not in short videos. Interestingly there was no behavioral effects with priming. The study is written well and the experiments are well designed. Their statistical analysis on ERP effects was appropriate and well done. The findings would be contributory to the literature. I have a few comments/suggestions/questions.
What were the pseudowords used in this study? This was not explained in the methods. I think their experiment was well designed, but they did not include ERP data associated with pseudowords. Contrasting words vs pseudowords may provide a reference point (possibly a more typical N400 effect) to compare how their P300 findings may differ in time and distribution from simply getting access to lexico-semantic information (by contrasting words vs pseudowords). This is usually the benefit of having a lexical decision design. Also this would provide important theoretical implications of their overall findings.
Why did they choose to use block design? Based on what I read they put long duration video primes and short duration primes in separate blocks, rather than a mixed design where conditions would be intermixed within the same blocks. One issue I can imagine is that perhaps the blocks with long duration would take more time and the participants may need to concentrate more, or they may be more engaged and paying more attention. Could these factors simply explain the effects they found because of the block design they used?
400 ms video clips seem quite brief. Did they test if these videos were recognizable for those actions they represent? As opposed to the shallow processing vs deep processing that contributed to the findings, could it simply be that 400 ms video clips were more ambiguous, such that actions between clear/drop/smash were not easily distinguished with 400ms presentation? This may provide an alternative explanation to their discussion in 4.2, suggesting that these ambiguous action videos may not generate "specific" semantic activation of action concepts, but rather general, leading to no congruency effects. However, if this is the case, it would almost indicate that replacing these short videos with any other actions (such as playing football, table tennis, etc.) would not change the results. It would thus be important again to test if different actions can be recognized by watching these short videos. whether short videos would not elicit its corresponding action concept (without conscious semantic activation), or can actually elicit action concepts but with shallower processing (with shallower semantic activation), may lead to very different interpretations of their results.
It is not clear enough to me during the experiment "to match the processing difficulty of the videos among the four conditions" (line 141), did that independent group also performed a lexical decision task? If not what did they do? It seems that with this independent group of subjects with similar age range had slower RT (around 750 ms) and worse accuracy (around 85%) based on the means, compared to the full group (RT around 430 ms and accuracy around 93%). How do they explain this if their goal was to find a group with comparable performance.
Would need to provide rationale for why P300 and LPC analysis were focused on 200-400 ms and 500-700 ms windows (but not for example, 200-600 ms and 600-900ms, that may also be chosen by other studies)?
I would advise caution in linking action processing (based on video watching) to motor system, as mentioned a few times in their discussion (line 350, 409). They did not directly test motor systems in the study and the ERP effects are not directly associated with motor systems given unclear source generation. This point has need to be made clear.
It is unclear to me what it means when LPC represents retrieval of episodic information (starting from line 397). Does this mean that participants remember seeing these videos that were presented previously in the same block that somehow may be retrieved by reading the word stimuli? Or what kind of "episodic memories" did they mean? Alternatively, there is definitely a wealth of literature on LPC associated with semantic/linguistic processing (https://pubmed.ncbi.nlm.nih.gov/?term=%28semantic+processing%29+AND+%28late+positive+component%29). Would that also be possible that long videos may elicit deeper processing for words regardless of congruency effects?
Round 2
Reviewer 1 Report
The authors addressed the issues raised and the paper improved.
Reviewer 2 Report
I'm content with the authors' responses. I have no more comments.